# Unraveling the dislocation core structure at a van der Waals gap in bismuth telluride

D.L. Medlin[1], N. Yang[1], C.D. Spataru[1], L.M. Hale[2] & Y. Mishin[3]

Tetradymite-structured chalcogenides such as bismuth telluride ($Bi_2Te_3$) are of significant interest for thermoelectric energy conversion and as topological insulators. Dislocations play a critical role during synthesis and processing of such materials and can strongly affect their functional properties. The dislocations between quintuple layers present special interest since their core structure is controlled by the van der Waals interactions between the layers. In this work, using atomic-resolution electron microscopy, we resolve the basal dislocation core structure in $Bi_2Te_3$, quantifying the disregistry of the atomic planes across the core. We show that, despite the existence of a stable stacking fault in the basal plane gamma surface, the dislocation core spreading is mainly due to the weak bonding between the layers, which leads to a small energy penalty for layer sliding parallel to the van der Waals gap. Calculations within a semidiscrete variational Peierls-Nabarro model informed by first-principles calculations support our experimental findings.

[1] Sandia National Laboratories, Livermore, CA 94551, USA. [2] Materials Measurement Laboratory, National Institute of Science and Technology, Gaithersburg, MD 20899, USA. [3] Department of Physics and Astronomy, MSN 3F3, George Mason University, Fairfax, VA 22030, USA. Correspondence and requests for materials should be addressed to Y.M. (email: ymishin@gmu.edu)

Layered, tetradymite-structured chalcogenides are of tremendous technological interest due to the novel electronic and thermal transport properties that are imparted by their quasi- two-dimensional (2D), sheet-like structures. In such structures, thin sheets composed of several atomic layers weakly interact with each other by van der Waals forces across interlayer regions called van der Waals gaps. These materials have long been of great interest as thermoelectrics[1–7] and, more recently, have been of intense focus in the context of topological insulators[8–12]. It is important to understand the nature of extended crystallographic defects, such as dislocations, in these materials. Dislocations are 1D topological defects in crystalline materials whose glide in certain crystallographic planes constitutes the main mechanism of plastic deformation[13]. A typical dislocation consists of a core region with large atomic displacements from perfect-lattice positions, and an elastic strain field extending deep into the surrounding crystal lattice. Dislocations in the layered chalcogenides are relevant to both the processing and functional properties of these materials. Polycrystalline bulk thermoelectrics have long been processed by thermomechanical means that introduce dislocations through plastic deformation in order to improve their densification and control their crystallographic texture[14]. Dislocations are also relevant in the context of chalcogenide nanostructures and epitaxial films, for which growth spirals and low-angle grain boundaries associated with threading dislocations are commonly reported[15–17].

Dislocations in quasi-2D chalcogenides can also strongly affect functional properties of these materials by several mechanisms. For example, recent work has suggested that dislocations present at low-angle grain boundaries in $Bi_2Te_3$-based alloys effectively scatter phonons in the mid-frequency range[18,19], providing a grain-boundary design strategy for engineering new materials with improved thermoelectric energy conversion efficiency. The core structure of the dislocations affects their mobility under applied mechanical stresses and thus the dislocation density in the microstructure, as well as the effectiveness of phonon scattering by individual dislocations. The large strain fields near dislocation cores can also affect the electronic band structure. For instance, scanning tunneling microscopy measurements near low-angle grain boundaries in $Bi_2Se_3$ thin films grown by the molecular beam epitaxy method have demonstrated shifts in the energy of the Dirac state, which were attributed to the large strain fields near the individually resolved dislocation cores[15]. Such strain fields are characterized by alternating tension and compression regions. Experiments and first-principles calculations[5,20] have shown that application of tensile and compressive strains across the van der Waals gap in $Bi_2Se_3$ strongly affects the electronic band structure. Tensile strain was found to shift the Dirac point, while compressive strain opened a gap and destroyed the Dirac states. These observations suggest that electronic and other functional properties of the material can be strongly impacted by the core structure of dislocations present at the van der Waals gap. The motion and interaction of dislocations under thermomechanical processing can also strongly alter the carrier concentrations, which directly affects properties such as the electrical resistivity and thermopower[21–23].

A central question that remains unresolved is how the weak bonding across the van der Waals gap affects the atomic structure of the dislocations present at the interlayer. This is important since the structure of a dislocation is central to its properties and behavior. For instance, the local elastic strain distribution as well as the specific atomic arrangements in the vicinity of a dislocation core depends on whether the core is compact or dissociated into partial dislocations, which directly impacts the local electronic states and behavior of charge carriers[24,25]. In this paper, we determine the structure of dislocations present at the basal plane of $Bi_2Te_3$, a prototypical tetradymite-structured compound. Our approach is a combination of experimental observations employing atomic-resolution electron microscopy, and computer modeling encompassing ab initio electronic structure calculations and continuum/discrete dislocation theory. The local structural disruption imposed by a dislocation presents us with an opportunity to probe the interaction strength across the interlayer gap. Our measurements of the dislocation core spreading, analyzed using a semidiscrete Peierls–Nabarro framework, provide fundamental insights concerning the dislocations in $Bi_2Te_3$ and, by extension, other layered materials. This work additionally allows us to compare the predictive capabilities of several commonly used exchange-correlation functionals incorporating van der Waals corrections into the density functional theory (DFT) calculations. The approaches described here are generally applicable and provide a path to a deeper understanding of extended defect structures and their connection to interlayer bonding in this important class of materials.

## Results

**Experimental determination of the dislocation core structure.** The crystal structure of $Bi_2Te_3$ is illustrated in Fig. 1a. The material has a rhombohedral crystal structure ($R\bar{3}m$, $a = 0.438$ nm, $c = 3.05$ nm)[26–28] consisting of hexagonal sheets of alternating bismuth and tellurium atoms stacked along the $c$-axis in 5-plane groupings, or quintuple layers, of the form $Te^{(1)}$-$Bi$-$Te^{(2)}$-$Bi$-$Te^{(1)}$. Three of such quintuple layers constitute a single unit cell. The van der Waals gap is parallel to the basal plane and corresponds to the region between abutting $Te^{(1)}$:$Te^{(1)}$ planes. The material is isomorphous with $Sb_2Te_3$ and $Bi_2Se_3$, which are typically alloyed with $Bi_2Te_3$ for thermoelectric applications[4], and which are of interest in their own right as topological insulators[10]. All these materials fall within the broader class of tetradymite-type compounds[29], which possess similar layered structures.

Previous work has identified two primary types of dislocation in $Bi_2Te_3$ and the isomorphous compounds: dislocations with Burgers vector, **b**, of type $\frac{1}{3}\langle 2\bar{1}\bar{1}0 \rangle$[30–34], which lies parallel with the basal plane (Fig. 1b, c), and those with **b** of type $\frac{1}{3}\langle 0\bar{1}11 \rangle$[17,35], which has a large component normal to the basal plane. The Burgers vector **b** characterizes the magnitude and direction of the lattice translation produced by the dislocation. Operationally, it is given by the closure failure, when mapped onto the perfect lattice, of an imaginary closed loop constructed around the region including the dislocation (the so-called Burgers circuit)[13]. We focus in this paper on the **b** = $\frac{1}{3}\langle 2\bar{1}\bar{1}0 \rangle$-type dislocations as their structure is directly related to the fundamental interactions across the interlayer gap. Since this Burgers vector is parallel to the basal plane, the dislocations can glide on this plane under applied shear stresses.

The presence of $\frac{1}{3}\langle 2\bar{1}\bar{1}0 \rangle$ dislocations in $Bi_2Te_3$ has long been recognized. In fact, the pioneering electron microscopy study of $\frac{1}{3}\langle 2\bar{1}\bar{1}0 \rangle$ dislocation networks in $Bi_2Te_3$ and $Sb_2Te_3$ by Amelinkx and Delavignette[30] and Delavignette and Amelinkx[31] provided one of the first direct observations of dislocations in non-metallic materials. The assumption has long been that the cores of such dislocations would be localized at the van der Waals gap between the $Te^{(1)}$:$Te^{(1)}$ planes, where the bonding is weakest. Indeed, in situ observations of gliding dislocations in $Bi_2Te_3$-based alloys provide some evidence supporting this assertion[32,33]. Recently, Fu et al.[34] have shown that the cores of $\frac{1}{3}\langle 2\bar{1}\bar{1}0 \rangle$ dislocations in $Sb_2Te_3$ are localized in the $Te^{(1)}$:$Te^{(1)}$ layers. However, direct proof of this localization in $Bi_2Te_3$ or detailed knowledge of the core structure of such dislocations have been missing. It was hypothesized[31] that the core was dissociated into $\frac{1}{3}\langle 0\bar{1}10 \rangle$-type Shockley partials by analogy with similar dissociation taking place

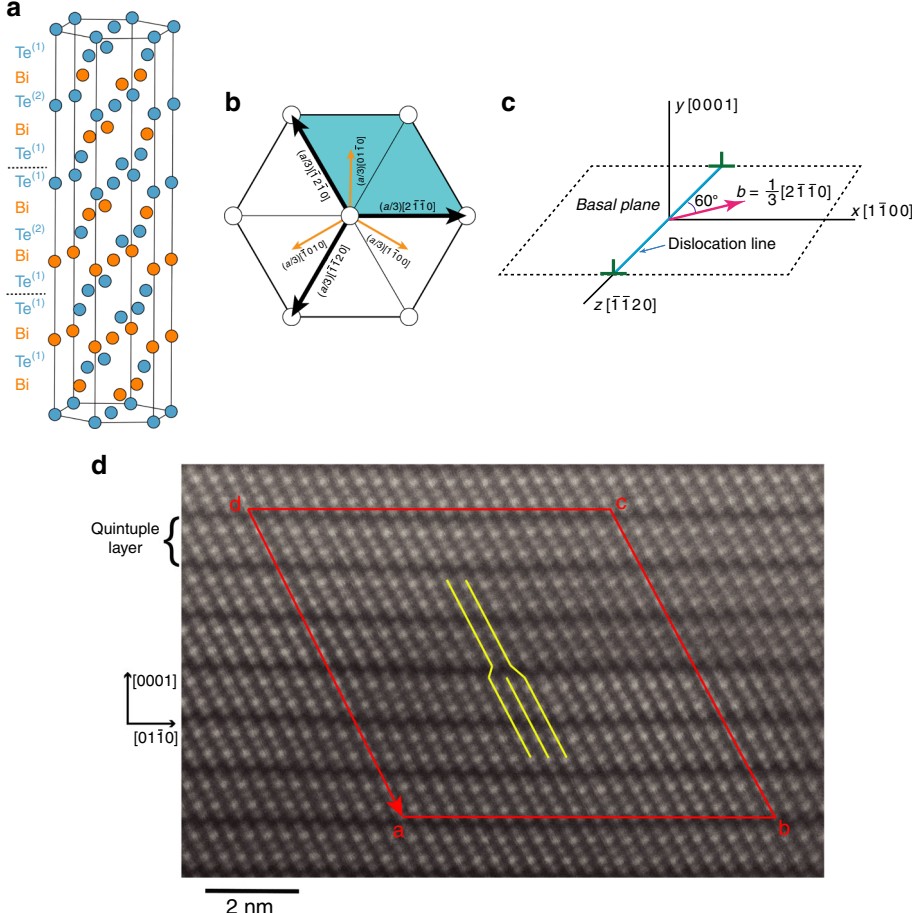

**Fig. 1** Crystallographic details of $Bi_2Te_3$ and the dislocation. **a** Atomic arrangements in $Bi_2Te_3$. The space between the $Te^{(1)}$:$Te^{(1)}$ planes is the van der Waals gap. **b** Projection of the structure on the basal plane showing the Burgers vectors of dislocations. The unit cell is shaded in blue. The heavy black arrows show the Burgers vectors for the $\frac{1}{3}\langle 2\bar{1}\bar{1}0\rangle$ -type perfect-lattice dislocations, whereas the smaller orange arrows show the $\frac{1}{3}\langle 10\bar{1}0\rangle$ -type Burgers vectors that would result if Shockley partial dislocations were to form. **c** Orientation of the Cartesian axes relative to the dislocation line. **d** High-resolution transmission electron microscopy (HRTEM) image of $Bi_2Te_3$ projected along $\langle 2\bar{1}\bar{1}0\rangle$ direction, showing the quintuple layers and the Burgers circuit construction for calculation of the dislocation Burgers vector (see Supplementary Note 2 for detail). The basal planes are horizontal and the yellow lines trace $\{10\bar{1}5\}$ crystal planes, one of which terminates at the dislocation core. The scale bar represents 2 nm

in close-packed metals. However, the microscope resolution at the time did not permit the authors to test this hypothesis. As demonstrated in the present paper, the suggested dissociation into partials does not occur in $Bi_2Te_3$.

To directly determine the core structure of the $\frac{1}{3}\langle 2\bar{1}\bar{1}0\rangle$ dislocations, we conducted an electron microscopic study of a polycrystalline sample of $Bi_2Te_3$ consolidated through a thermo-mechanical processes expected to produce dislocations. We employed the technique of high-angle annular dark-field (HAADF) scanning transmission electron microscopy (STEM), using a probe-corrected instrument operated at 300 keV (see Methods for further details). This technique allows the Te and Bi layers to be distinguished due to their large difference in atomic number[36]. Figure 1d shows an atomically resolved HAADF-STEM image of a dislocation observed in $Bi_2Te_3$. From Burgers circuit analysis, we confirm that the dislocation is of $\frac{1}{3}\langle 2\bar{1}\bar{1}0\rangle$ type. The image itself is projected along a $\langle 2\bar{1}\bar{1}0\rangle$-type orientation. Assuming that the line direction of the dislocation is along this projection, the Burgers vector is inclined by ±60° with respect to the dislocation line. Thus, the dislocation is of 60° mixed type with both edge and screw components. Inspection of the image shows clearly that the dislocation core terminates at the $Te^{(1)}$:$Te^{(1)}$ layer as expected. Note that the $Bi_2Te_3$ quintuples above and

below the slip plane remain intact, suggesting that the dislocation core maintains stoichiometry. We have analyzed a total of six different dislocations of this type, all of which terminated in the same manner. This can be contrasted to $\frac{1}{3}\langle 0\bar{1}11\rangle$ edge dislocations, which have been observed to form a dissociated, Bi-rich core[35].

The question arises as to how localized the dislocation core is within the glide plane. The degree of dissociation is absolutely central to the properties and behavior of dislocations (e.g., controlling the ease of deformation processes such as cross-slip). The dissociation width depends on the shape of the so-called gamma-surface, which is the excess interlayer energy $\gamma$ as a function of the translation vector **t** parallel to the layers[37–39]. The gamma-surface, in turn, is sensitive to the strength of the interlayer interactions. Thus, measuring the core dissociation and reconstructing the respective gamma-surface provides an effective way of assessing the character of interlayer interactions.

To accomplish this, we measured the disregistry $\delta$ of the atomic planes at the dislocation slip plane from the positions of $\{10\bar{1}5\}$ planes within the quintuple units above and below the slip plane. The atomic-plane positions on either side of the slip plane were determined by the template averaging and matching method illustrated in Fig. 2a and described in more detail in the Methods

**a**

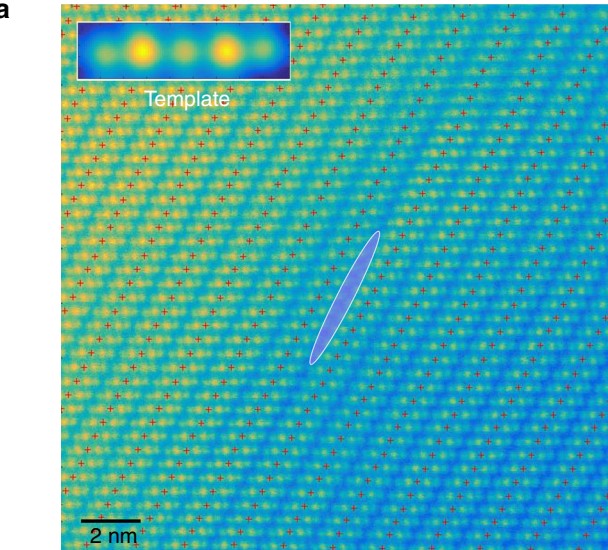

**b**

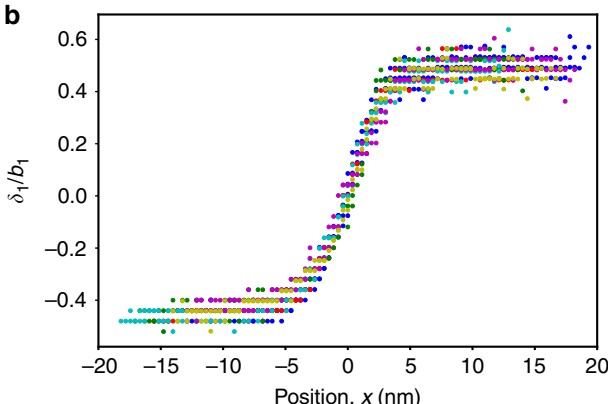

**c**

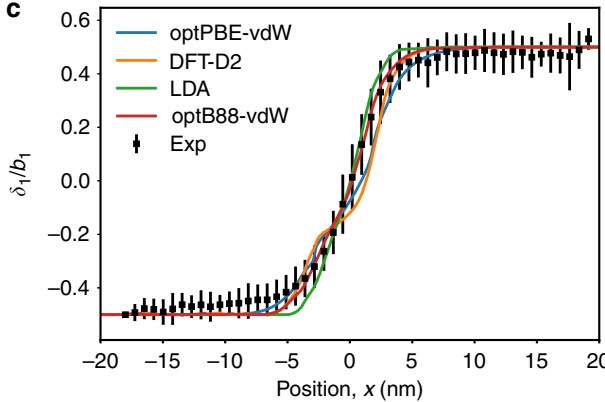

**Fig. 2** Disregistry at the dislocation core. **a** Illustration of the templating procedure for extracting the atomic-plane disregistry in the dislocation core, with the cross symbols indicating the centers of the quintuple structural units on either side of the slip plane. The dislocation core is highlighted in purple. The scale bar represents 2 nm. **b** Disregistry as a function of distance $x$ across the dislocation core for six (color-coded) dislocations observed in this work. **c** Disregistry as a function of distance $x$ averaged over the six dislocations and compared with predictions of the semidiscrete Peierls–Nabarro model. The experimental disregistry $\delta_1$ has been normalized by the edge component $b_1$ of the experimental Burgers vector. The error bars represent two standard deviations. The curves were obtained using different density functional theory (DFT) functionals indicated in the legend

section. The disregistry, $\delta = (u_+ - u_-)$, is calculated from the intersections of the $\{10\bar{1}5\}$ planes with the slip plane. Here, $u_+$ is the intersection extrapolated from the planes above the slip plane and $u_-$ is the intersection extrapolated from below the slip plane. The collection of disregistry plots $\delta(x)$ measured from the six different dislocations is displayed in Fig. 2b. The $x$-direction is parallel to the edge component of the Burgers vector. The results were combined into a single plot by binning along the $x$-axis and computing the average and standard deviation of the disregistry in each bin (Fig. 2c). The striking feature of the disregistry plot is that it shows a rather wide (over 1 nm) dislocation core, yet no signs of dissociation into well-defined partials, contrary to what was suggested in ref. [31].

**Calculation of the dislocation core structure**. To understand the nature of the unusually wide and yet undissociated dislocation core, the gamma-surface on the basal plane was computed by first-principles DFT methods. The challenge of this calculation was that different exchange-correlation functionals available in the literature produce qualitatively different shapes of the gamma-surface, which in turn leads to different predictions of the detailed core structure. To arrive at definitive conclusions, several different DFT functionals were tested, including the local-density approximation (LDA)[40], the non-local correlation functionals optPBE-vdW and optB88-vdW[41–44] accounting for dispersion interactions, and the DFT-D2 functional[45] introducing semi-empirical van der Waals corrections (see Methods for details). The gamma-surface was computed with each of these functionals (Fig. 3a). All functionals predict the existence of a stable stacking fault (SF) on the basal plane, which is obtained by a relative translation of the $Te^{(1)}$ layers across the van der Waals gap. An SF is a planar (2D) defect of a crystal structure obtained by relative translation of two half-crystals to create the wrong stacking sequence of the crystal planes. While the translation vector corresponding to the local minimum of the fault energy $\gamma_{sf}$ remains approximately the same, the depth of the minimum and the respective SF energy depend on the functional. The LDA approximation predicts a large SF energy and a shallow minimum, whereas the DFT-D2 functional predicts the lowest SF energy with a broad minimum surrounded by relatively high barriers.

Based on the computed gamma-surface $\gamma(\mathbf{t})$, the disregistry function $\delta(x)$ in the dislocation core region can be predicted theoretically using one of the Peierls–Nabarro-type models[13,46]. The classical Peierls–Nabarro model represents an edge dislocation core by a continuous function $\delta(x)$ satisfying the boundary condition $\delta(\infty) - \delta(-\infty) = b$, where $b$ is the magnitude of the Burgers vector. It additionally postulates a sinusoidal shape of the gamma-surface in the $x$-direction, which is not suitable for our purposes. Instead, we employed a semidiscrete variational Peierls–Nabarro (SDVPN) model[47–49], which was properly generalized in this work to capture the elastic anisotropy of the $Bi_2Te_3$ crystal structure. In this model, the disregistry $\delta_i$ is a vector (with three components labeled by index $i$) and is evaluated at discrete values of $x$ equally spaced by $\Delta x$. The latter corresponds to a spacing of atomic columns parallel to the $x$-direction. The model can be applied to an arbitrary (mixed-type) dislocation. The dislocation core is represented by a discrete set of imaginary partial dislocations running parallel to the actual dislocation line. The core energy is the sum of the elastic strain energy arising from the interaction of the partial dislocations, plus the crystal misfit energy due to the core spreading. The latter contribution depends on the shape of the gamma-surface provided as input to the model. In the variational formulation of the model, the disregistry function $\delta_i(x)$ is found by minimizing the total energy.

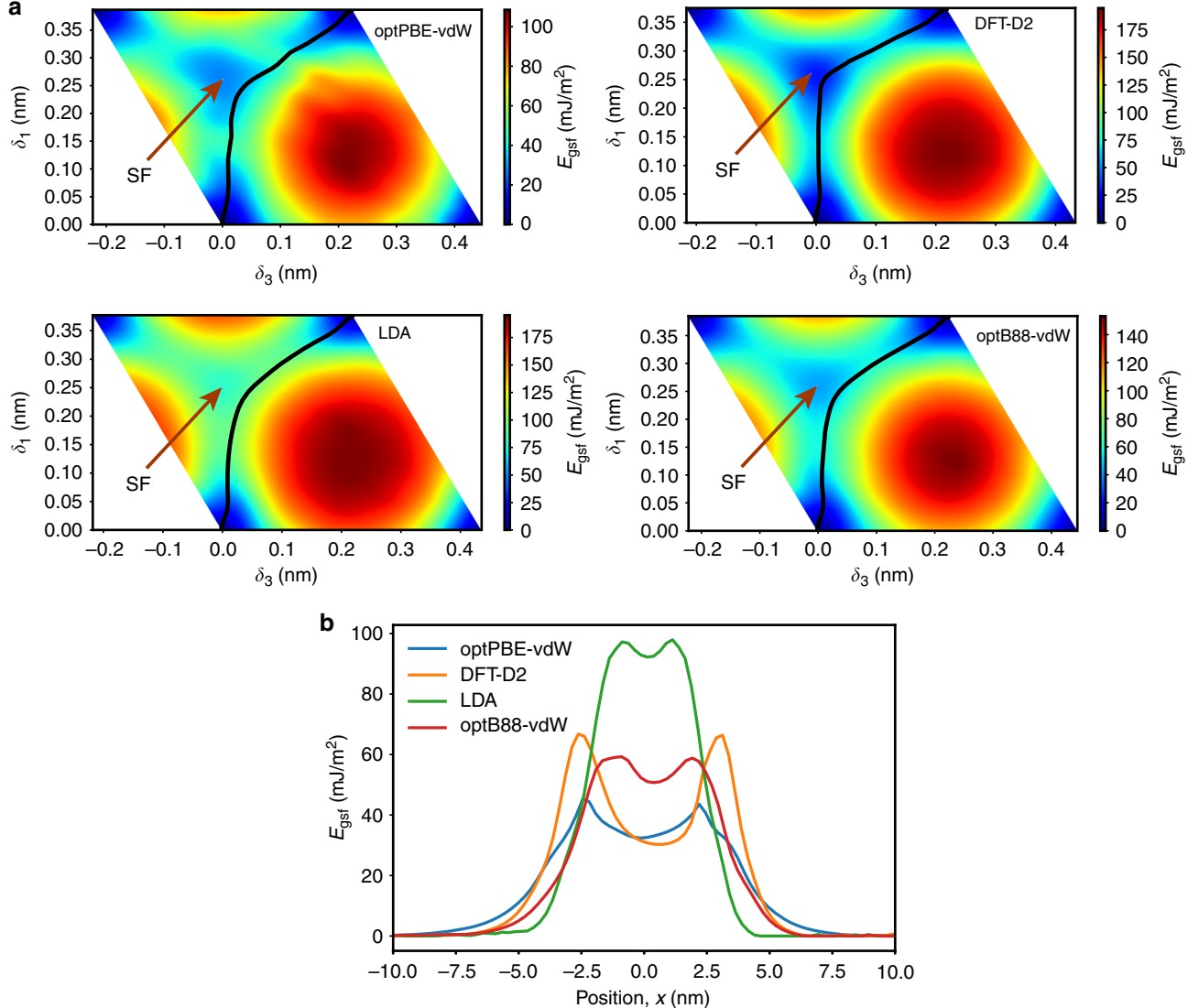

**Fig. 3** The gamma-surface and the stacking fault (SF). **a** Gamma-surfaces computed with different density functional theory (DFT) functionals. The generalized SF energy $E_{gsf}$ is only shown within a repeat unit parallel to the basal plane. The stable SF position (local minimum on the gamma-surface) is indicated. The black line shows the disregistry path within the dislocation core region predicted by the semidiscrete Peierls–Nabarro model using the respective gamma-surface. **b** The generalized SF energy as a function of distance x across the dislocation core computed with different DFT functionals. Note that, with the exception of the DFT-D2 functional, all other DFT calculations predict remarkably low barriers (on the order of $10\,mJ\,m^{-2}$) on either side of the local minimum representing the stable SF

Further details of the model are described in the Supplementary Note 1.

## Discussion

Figure 2c compares the SDVPN model predictions with the experimental disregistry function. We emphasize that the SDVPN calculations did not involve any fitting to experimental data. The immediate conclusion is that the model predicts the core shape in very reasonable agreement with experiment, confirming that the wide spread of the dislocation core is primarily due to the weak van der Waals bonding across the gap, not a result of dissociation into two discrete partials separated by a SF ribbon as in low SF energy face-centered cubic (FCC) metals. In the latter case, the disregistry function would have a well-pronounced flat region at the center of the core, which is not observed in Fig. 2c. This conclusion is additionally supported by the disregistry trajectories (Fig. 3a), showing the edge ($\delta_1$) and screw ($\delta_3$) components. The

trajectories pass close to the SF position but never reach it, showing that a real SF ribbon does not form. (A similar situation with very narrow dissociation without the formation of a real SF ribbon is found in high SF energy FCC metals such as Al[50], but then the dislocation core is much more compact than in Bi$_2$Te$_3$). A closer inspection of Fig. 2c reveals that the different exchange-correlation functionals utilized for the DFT gamma-surface calculations lead to slightly different shapes of the disregistry plots. The LDA approximation underestimates the core width, which is consistent with the prediction of the high SF energy (cf. Figure 3b). The optPBE-vdW and especially DFT-D2 functionals create a kink at the center of the core that is not supported by the experimental disregistry function within the statistical error bars. This kink arises from the relatively low SF energy and the wide separation of the energy barriers around the energy minimum (cf. Figure 3b). Thus, an additional outcome of the present calculations is that they provide a useful benchmark of the DFT functionals, pointing to optB88-vdW as an accurate model for van der

Waals materials such as $Bi_2Te_3$. The SF energy $\gamma_{sf}$ predicted by this functional is 46 mJ m$^{-2}$ (Supplementary Table 1).

These results indicate that the wide spreading of the dislocation core is primarily caused by the weak van der Waals bonding between the quintuple layers. This weak bonding results in the relatively low unstable SF energy $\gamma_{us} \approx 60$ mJ m$^{-2}$ (the maxima in Fig. 3b) compared to 65–200 mJ m$^{-2}$ or higher in FCC metals[51–54]. In other words, the penalty for the translations of the layers parallel to the van der Waals gap in $Bi_2Te_3$ is relatively small. Perhaps, more importantly, the depth ($\gamma_{us} - \gamma_{sf}$) of the local energy minimum corresponding to the stable SF in $Bi_2Te_3$ is unusually small, about 10 mJ m$^{-2}$. For comparison, in Cu $\gamma_{sf}$ is about the same as in $Bi_2Te_3$ but $\gamma_{us}$ is about 160–180 mJ m$^{-2}$[51,54,55]. As a result, the stable SF minimum is surrounded by much higher (about 100 mJ m$^{-2}$) energy barriers. These high barriers lead to the dissociation of the dislocation core structure into cleanly separated, discrete Shockley partials. As another comparison, the so-called MAX phases[56] also have a hexagonal layered structure, but the bonding between the layers is not van der Waals type. Unlike in $Bi_2Te_3$, the full dislocations on the basal plane dissociate into discrete Shockley partials separated by a SF[57]. The energy minimum corresponding to the SF is relatively shallow, but the absolute values of all energies on the gamma-surface are at least an order of magnitude higher than in $Bi_2Te_3$, including the minimum depth ($\gamma_{us} - \gamma_{sf}$). Similarly, in Ti the prismatic SF along the $\frac{1}{3}\langle 11\bar{2}0\rangle$ direction has a relatively shallow minimum, but the absolute values of all energies are significantly higher than in $Bi_2Te_3$[58–60]. The unique feature of the basal dislocations in $Bi_2Te_3$, and most likely in other tetradymite-structured chalcogenides, is the combination of the low unstable SF energy with the shallowness of the stable SF minimum. This combination prevents any significant localization of the dislocation content into partial dislocations. Even if such dislocations hypothetically formed, their cores would be wide and would strongly overlap, making the very concept of dissociation meaningless. Thus, despite the existence of a stable SF in the gamma-surface, the wide spreading of the dislocation core in $Bi_2Te_3$ is not accompanied by dissociation into two discrete partials. The core remains wide but undissociated. This important conclusion is supported by calculations within a semidiscrete Peierls–Nabarro model with input from DFT calculations, showing excellent agreement with experiment.

In summary, we have been able to resolve the detailed basal dislocation core structure in the layered chalcogenide $Bi_2Te_3$, and to quantify the atomic-plane disregistry across the dislocation core using atomic-resolution electron microscopy. The wide spreading of the dislocation core is mainly due to the weak, van der Waals type bonding between the quintuple layers of the structure, which leads to relatively low energy barriers for the SF formation and annihilation. As an additional finding, we have identified the exchange-correlation functional (optB88-vdW) that should be most appropriate for future dislocation modeling in $Bi_2Te_3$ and, by extension, other materials composed of van der Waals bonded atomic layers[61]. As discussed in the beginning of the paper, the dislocation core structure in $Bi_2Te_3$ can impact many functional properties of the material. As additional evidence, Supplementary Note 3 reports on preliminary DFT calculations of the electronic properties for an idealized SF in $Bi_2Te_3$. The results suggest that the SF is likely to affect the concentration of free charge carriers in the SF region. A more detailed analysis of the dependence of the electronic structure on the variation of the disregistry on the basal plane would be a useful future step toward predicting the electronic effects of the actual basal plane dislocations in $Bi_2Te_3$ and related materials that exhibit a spread core.

## Methods

**Experimental methodology.** The observations were conducted on polycrystalline $Bi_2Te_3$ material that had been initially consolidated by spark-plasma sintering and then further processed by hot extrusion. This extrusion technique is convenient for the present study since the deformation introduces dislocations and can drive a preferential crystallographic texture of $\langle 2\bar{1}\bar{1}0\rangle$[62], which is an ideal imaging direction for atomically resolving the dislocation core and its displacements. The sintering employed procedures detailed in ref. [63]. Following sintering, the $Bi_2Te_3$ puck was subjected to extrusion to increase its density. During preparation, the puck was coated with a high temperature graphite lubricant to ensure smooth movement through the tooling. The tooling was heated to 400 °C and held at this temperature for 30 min to enhance $Bi_2Te_3$ plasticity during the extrusion treatment. The puck was then pressed through a tool steel die with a reduction ratio of 4:1 using a force of ~$10^4$ kg at a strain rate of 0.1 s$^{-1}$.

The $Bi_2Te_3$ specimen for electron microscopy analysis was thinned to electron transparency by mechanical grinding and dimpling, followed by Ar$^+$ ion milling using a Fischione Model 1010 ion milling system. The specimen was cooled during ion milling using a liquid nitrogen stage. Analysis was conducted by HAADF-STEM using a probe-corrected 80-300 FEI Titan instrument operated at 300 keV. Images were collected with the local grain region oriented along a $\langle 2\bar{1}\bar{1}0\rangle$-type direction. In order to efficiently identify dislocations over a large field of view, initial imaging was conducted at low magnification with a scan sampling frequency and scan orientation selected to provide strong Moiré contrast from the dislocation. By imaging at low magnification, we also took care to ensure that the analyzed dislocations were chosen to be well separated from grain boundaries and other dislocations by a radius of at least 75 nm. After identifying and focusing on each dislocation from its Moiré image, higher magnification, atomic-resolution images were collected. Image analysis was conducted using ImageJ and custom routines written in MATLAB.

The disregistry across the dislocation slip plane was measured from the atomically resolved HRSTEM images. A total of six dislocations were analyzed from images with a sampling of 77 pixels/nm. Multiple images were analyzed for each dislocation (four images for one of the dislocations; two images each for the remaining five dislocations). We employed a template averaging and matching approach[64] to determine the atomic-plane positions on either side of the slip plane. We then computed the difference in these positions as a function of distance to determine the disregistry. An example illustrating the template matching approach is shown in Fig. 2a. In detail, the images were first digitally rotated to align the average {10$\bar{1}$5} plane orientation parallel with the horizontal image axis. Next, an initial, trial template region was defined by selecting a rectangular region of the image encompassing a single Te$^{(1)}$-Bi-Te$^{(2)}$-Bi-Te$^{(1)}$ unit along a {10$\bar{1}$5} plane. The trial template was then cross-correlated with the starting image. Image patches centered on the peaks in the cross-correlation function were then averaged to give a refined template pattern. Finally, the refined template pattern was cross-correlated with the starting image, and the peaks from this function were extracted to determine the atomic motif positions, and hence the local position {10$\bar{1}$5} planes on either side of the slip plane.

The disregistry at the slip plane was determined by projecting the {10$\bar{1}$5} planes from above and below the slip plane to the mid-plane between Te$^{(1)}$-Te$^{(1)}$ layers and calculating the distance between these intersections. Since the {10$\bar{1}$5} planes are aligned parallel with the $x$-axis of the analysis coordinate frame, the disregistry, $\delta = (u_+ - u_-)$, was calculated for each plane intersection from the equation

$$u_+ - u_- = (y_+ - y_-)\sqrt{\frac{1 + m^2}{m^2}}, \qquad (1)$$

where $y_+$ and $y_-$ are the $y$-coordinates of the motifs measured on the two sides of the slip plane. $m$ is the slope of the mid-plane line calculated by fitting lines to the motif positions on the sides of the slip plane and taking the average slope for these two lines.

**Simulation methodology.** First-principles DFT calculations were performed using projector-augmented pseudopotentials as implemented in the electronic structure Vienna Ab initio Simulation Package[65,66]. Gamma-surface calculations were performed using 6-quintuplet slabs, with changes in total energy obtained after translating the upper 3 quintuplets parallel to the hexagonal basal plane. The translation vector **t** sampled the lateral area of the conventional unit cell on a uniform $10 \times 10$ mesh. After each translation, the atomic positions were relaxed in the direction normal to the fault. The lateral lattice vectors (in Cartesian coordinates) $\mathbf{a}_1 = a(1, 0, 0)$ and $\mathbf{a}_2 = (a/2)(\sqrt{3}, 1, 0)$ and the initial atomic positions were obtained for the lattice parameter $a$ computed by relaxing forces and stresses for bulk $Bi_2Te_3$. Convergence with respect to the Brillouin zone sampling was achieved employing uniform Monkhorst–Pack $k$-point meshes with sizes up to $7 \times 7 \times 1$. A smooth function $\gamma(\mathbf{t})$ was obtained by interpolating between the measured points using a multiquadric radial basis function. Several exchange-correlation functionals were employed, namely, LDA[40], optPBE-vdW and optB88-vdW[41–44], and DFT-D2[45]. The lattice parameters and elastic constants obtained with these functionals are summarized in Supplementary Table 1 and are in good agreement with previous DFT calculations[67].

In the SDVPN model used here, the disregistry function $\delta_i(x)$ was discretized on a mesh $\{x^\alpha\}_{\alpha = 1,...,N}$ of $N = 300$ points with the spacing $\Delta x = a\sqrt{3}/3$. The total

dislocation energy is $E_{total} = E_{elastic} + E_{misfit}$, where the elastic strain energy $E_{elastic}$ depends on the energy coefficients $K_{ij}$, which in turn depend on the elastic constants $C_{ij}$. The misfit energy $E_{misfit} = \sum_{\alpha=1}^{N} \gamma[\delta_i(x^\alpha)]\Delta x$ was obtained from the DFT gamma-surface. The equilibrium disregistry $\delta_i(x^\alpha)$ was found by numerical minimization of $E_{total}$ under appropriate boundary conditions. As the initial guess, we used the arctangent disregistry function predicted by the classical Peierls–Nabarro model. A continuous disregistry curve was obtained by smooth interpolation between the $x^\alpha$ points. The calculations were performed separately for each of the DFT functionals. Further technical details of the SDVPN calculations can be found in the Supplementary Note 1.

## Data availability

Calculations within the SDVPN model were implemented in Python and are openly available as part of the atomman Python package at https://github.com/usnistgov/atomman. All data that support the findings of this study are available in the Supplementary Information file or from the corresponding author upon reasonable request.

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

## Acknowledgements

Y.M. acknowledges support from the US Department of Energy, Office of Basic Energy Sciences, Division of Materials Sciences and Engineering, the Physical Behavior of Materials Program, through Grant No. DE-FG02-01ER45871. The research at Sandia National Laboratories was developed with funding from the Defense Advanced Research Projects Agency (DARPA) (C.D.S.) and the Laboratory Directed Research and Development program (D.L.M. and N.Y.). Sandia National Laboratories is a multimission laboratory managed and operated by National Technology and Engineering Solutions of Sandia, LLC, a wholly owned subsidiary of Honeywell International Inc., for the US Department of Energy's National Nuclear Security Administration under contract DE-NA0003525. This paper describes objective technical results and analysis. The views, opinions, and/or findings expressed are those of the authors and should not be interpreted as representing the official views or policies of the Department of Defense, Department of Energy, or the US Government. Certain commercial instruments, materials or processes are identified in this paper to adequately specify the experimental procedure. Such identification does not imply recommendation or endorsement by the National Institute of Standards and Technology, nor does it imply that the instruments, materials, or processes identified are necessarily the best available for the purpose.

## Author contributions

D.L.M. performed the TEM observations of the $Bi_2Te_3$ dislocations and measurements of the disregistry, and prepared a draft of the introduction and experimental part of the paper. N.Y. was responsible for processing the $Bi_2Te_3$ material used in this study. C.D.S. performed the DFT calculations and described the results and methodology, while L.M.H. conducted the calculations within the SDVPN model and prepared a draft of this part of the work. D.L.M. and Y.M. conceived this project and coordinated the collaborations among the co-authors. Y.M. produced an initial draft of the complete manuscript. All co-authors were engaged in discussions, contributed ideas at all stages of the work, participated in the manuscript editing, and approved its final version.

## Additional information

**Competing interests:** The authors declare no competing interests.

