## [Peer Review File · Nature Communications]

Reviewers' comments:

Reviewer #1 (Remarks to the Author):

This work deals with the observation of dislocation core structure in two-dimensional layered Bi₂Te₃ alloy, which is the state-of-the-art thermoelectric and topological insulating materials.

This reviewer is delighted to see the dislocation core structure at the basal plane of Bi₂Te₃ alloy that may be an important defect to understand the origin of physical properties. In the rigorous study on the structure in both experimental and theoretical methodologies, a good agreement between them support the main claim of the present work.

In my opinion, this paper can be published with minor additions that would be helpful to strengthen this work for a broad readership. The comments or theoretical calculations on the electronic structure or charge density map around dislocation core will be very interesting (or important), because the carrier generations of n-type and p-type is now understood by atomic defects in Bi₂Te₃ with Se and Sb alloying, respectively. If the dislocation core can have a significant role on the minor carrier generation, it is possible to provide a strategy for the control of carrier generations in the material preparation, leading to a high performance thermoelectric materials.

Can the author can comment on the possibility of the observed dislocation core structure as a general feature in 2D materials such as transition metal dichalcogenides or MAX phases??

Reviewer #2 (Remarks to the Author):

The authors present the core structure of a dislocation in layered Bi₂Te₃ using both HAADF-STEM and computations based on a generalized Peierls-Nabarro model. The model is informed by the generalized stacking fault energy surfaces as predicted by Density Functional Theory with several exchange correlation potentials. The major finding is that the full $\frac{1}{2}\langle 11-20 \rangle$ type dislocation does not dissociate into partials, but has a spread core due to weak van der Waals interaction between the layers. The results are interesting; however, I think the authors should clarify the following:

(1) On page 6, the authors argue that the reason for the spread core is the weak interlayer bonding. The most accurate exchange correlation potential, as determined by this study, predicts the SF energy to be 46 mJ/m². This value is surprisingly high and comparable to the SF energy in Cu. In the next paragraph, it is mentioned that the barrier to formation of this SF is 10 MJ/m² which is lower than typical FCC metals. My only interpretation is that the 10 MJ/m² is read from Fig3.b, where the stable stacking fault lies in a shallow minimum. The energy difference between the minimum and the neighboring maximum for the accurate functional (red curve) is about 10 mJ/m². If that's true, I don't think it is right to interpret this value as the energy barrier. The barrier to formation of the SF is from 0 to the maximum which is about 60 mJ/m². Since the dislocations are forming under shear, I don't think it's correct to think of the reverse mechanism (annihilation with a low barrier) as if this is a purely thermally activated process. In fact, the prismatic SF in Ti (along the $\frac{1}{3}\langle 11-20 \rangle$ has a similarly shallow minimum at the half point. I think this point needs to be sufficiently clarified as the main argument of the paper lies on the weak van der Waals interactions and it looks like the interactions are not that weak. Maybe they should be compared to the bonding within each layer.

(2) The introduction can provide more context about the impact of knowing the dislocation core structure. It is stated that dislocations can affect the functional properties of these important thermoelectric materials, however a step is missing to relate the core geometry to the functionalities. In my opinion, this should be emphasized for the Nature Communications audience.

Reviewer #3 (Remarks to the Author):

The manuscript by Medlin, Yang, Spataru, Hale and Mishin reports a combined experimental and theoretical study of the basal dislocation core structure in Bi₂Te₃. The authors combined first-principles DFT calculations with a semi-discrete dislocation theory model to obtain the theoretical results and compared with experimental observations employing scanning transmission electron microscopy, to arrive at the conclusion of the manuscript. The main result is that the spreading of the dislocation core is due to the weak bonding between the layers, leading to layer sliding parallel to the van der Waals gap. This study provides understanding toward an important observation.

However, I am unable to conclude that the present manuscript offers the sort of truly novel or unexpected findings that would likely to excite the immediate interest of a broad range of researchers. Unfortunately, it is currently not clear to me that it fits the criteria for publication in Nature Communications. These findings might be better suited for publication in a more specialized journal. I have a few technical questions and comments listed below, but my main concern for publication in Nature Communications is the lack of evidences that the manuscript will of interest to a broad research community.

1. The authors mention that "the large strain field near dislocation cores can also affect the electronic band structure." Have the authors investigated that with their DFT study? Can they provide any insight on that? Band structure calculations are fundamental to predict thermoelectric properties, which the authors mentioned as one of the main applications where their work can impact.
2. The authors mention that "the local structural disruption imposed by a dislocation presents us with an opportunity to probe the interaction strength across the interlayer gap." Can the authors provide any insight about the interaction strength in Bi₂Te₃ from their results?
3. How did the authors create dislocations in their DFT configurations? Did they relax the sample? Do the final results depend on the relaxation method used?
4. Did any input from the experimental data help with the theoretical modeling? It is not clear from the manuscript.
5. The manuscript is written assuming that the readers are familiar to the notations used, and not for a broad audience. The authors should include a brief description of the terminology they used in the main manuscript, to improve readability.
6. Why did the authors choose a polycrystalline sample? Could grain boundaries have an impact on their results?
7. How did they initiate the dislocation in the experimental sample? Is it present in the sample or the authors applied some shear? How likely it is that these dislocations occur in samples fabricated with advanced nanofabrication techniques, e.g. MBE? No such discussions are provided.
8. The authors mentioned that the results from their study can be generalized to understand other layered materials, but they did not provide any discussion. Without this kind of insight, unfortunately, the study seems too narrow in scope.

Response to Reviewers' comments

We are grateful to all three Reviewers for providing insightful comments on our paper. This document summarizes our responses to the Referees and the changes made in manuscript. Such changes include new DFT calculations suggested by the Reviewers, a number of modifications and additional comments in the main text of the paper that provide details and address the Reviewers' questions, 21 new literature citations, and a new section in the Supplementary Information file accompanied by 2 new figures.

Reviewer #1:

Reviewer:

In my opinion, this paper can be published with minor additions that would be helpful to strengthen this work for a broad readership. The comments or theoretical calculations on the electronic structure or charge density map around dislocation core will be very interesting (or important), because the carrier generations of n-type and p-type is now understood by atomic defects in Bi₂Te₃ with Se and Sb alloying, respectively. If the dislocation core can have a significant role on the minor carrier generation, it is possible to provide a strategy for the control of carrier generations in the material preparation, leading to a high performance thermoelectric materials.

Response:

This is a very good point. The goal of this work was to uncover the dislocation core structure in Bi₂Te₃. Its effect on the electronic structure and charge density presents interest but lies outside the scope of the paper. However, we agree with the Reviewer that pointing to the link between the dislocation core structure and the electronic properties of Bi₂Te₃ is important for putting this work in a broader context and reaching out to a broader readership. We note that the same suggestion was made by Reviewer #2 in his/her second comment, to which we provide a detailed answer below. We have rewritten the introductory part of the paper to discuss the literature data on the effects of the dislocation core structure in Bi₂Te₃ and other tetradymite-structured chalcogenides on functional properties of these materials. This includes the effect on the band structure, work function, carrier concentrations, thermopower, phonon scattering, etc. This discussion is accompanied by a number of literature citations. Furthermore, we have performed DFT calculations of charge density maps and electronic band structure for the SF region in comparison with the perfect material. Preliminary results of these calculations are included in the Supplementary Information file. (The calculations are limited to the SF structure because direct calculations of the widely spread dislocation core are not feasible for computational reasons.) We believe that these calculations, as well as the discussion of the literature data, provide a sufficient link to the functional properties. A more detailed analysis of this link is outside the scope of this paper and would only distract the attention from the main focus of this work.

Reviewer:

Can the author comment on the possibility of the observed dislocation core structure as a general feature in 2D materials such as transition metal dichalcogenides or MAX phases??

Response:

In the transition metal dichalcogenides, such as MoS₂, the situation is very different because the van der Waals bonding is weaker and the layers can easily bend. As a result, the dislocation cores are essentially surface ripples that feature different energetics and laws of interaction with each other than the traditional dislocations. To reflect these fundamental differences, these defects were termed “ripplocations” [Kushima et al, *Nano Lett.*, **2015**, 15 (2), pp 1302–1308]. In Bi₂Te₃ studied here, the quintuple layers are more rigid and the rippling effect is small, so the dislocations are conventional type. The MAX phases is a different story. They also have a hexagonal layered structure, but the bonding between the layers is not van der Waals type. Unlike in Bi₂Te₃, the full dislocations on the basal plane dissociate into discrete Shockley partials separated by a SF [Higashi et al. *Acta Mater.* **161** (2018) 161-170]. The energy minimum corresponding to the SF is shallow, but only in the relative sense, because all energies on the gamma surface are at least an order of magnitude higher than in Bi₂Te₃. This results in a different and more traditional dislocation core structure in comparison with Bi₂Te₃. A similar situation arises in HCP-Ti, where the SF on the prismatic plane corresponds to a relatively shallow minimum on the gamma surface (as pointed out by Reviewer#2). However, the bonding between the layers is not van der Waals type either, and stable and unstable SF energies are much higher than in Bi₂Te₃. These comments have been added in the penultimate paragraph of the paper.

Reviewer #2:

Reviewer:

(1) On page 6, the authors argue that the reason for the spread core is the weak interlayer bonding. The most accurate exchange correlation potential, as determined by this study, predicts the SF energy to be 46 mJ/M². This value is surprisingly high and comparable to the SF energy in Cu.

Response:

The 46 mJ/m² SF energy obtained in this work is indeed close to that in Cu. However, this SF energy can give rise to a widely dissociated core. With this SF energy, the SF ribbon in Cu is about 1 nm wide for the screw dislocation and about 2 nm wide for the edge dislocation (e.g., *Acta Mater.* **53**, 1313 (2005)). This is comparable to the width of the Bi₂Te₃ dislocation studied here and is significantly wider than the dislocation cores in high-SF energy metals such as Pt or Al.

Reviewer:

In the next paragraph, it is mentioned that the barrier to formation of this SF is 10 MJ/m² which is lower than typical FCC metals. My only interpretation is that the 10 MJ/m² is read from Fig3.b, where the stable stacking fault lies in a shallow minimum. The energy difference between the minimum and the neighboring maximum for the accurate functional (red curve) is about 10 mJ/m². If that's true, I don't think it is right to interpret this value as the energy barrier. The barrier to formation of the SF is from 0 to the maximum which is about 60 mJ/m². Since the dislocations are forming under shear, I don't think it's correct to think of the reverse mechanism (annihilation with a low barrier) as if this is a purely thermally activated process. In fact, the prismatic SF in Ti (along the 1/3<11-20> has a similarly shallow minimum at the half point. I think

this point needs to be sufficiently clarified as the main argument of the paper lies on the weak van der Waals interactions and it looks like the interactions are not that weak. Maybe they should be compared to the bonding within each layer.

Response:

We agree with the Reviewer that the language of this paragraph was not very clear. The barrier of the SF formation is indeed 60 mJ/m^2 and is given by the height of the maximum on the curve in Fig.3b. Another interpretation of this maximum is the unstable stacking fault energy γ_{US} . The 10 mJ/m^2 number mentioned in this paragraph is the depth of the shallow minimum corresponding the stable SF. This paragraph has been rewritten to clarify the two distinct features of the Bi_2Te_3 dislocation: (1) $\gamma_{\text{US}} = 60 \text{ mJ/m}^2$ is lower than γ_{US} is most FCC metals. For example, in Cu γ_{US} is between 160 and 180 mJ/m^2 (DFT calculations). Thus, the sliding of the quintuplet layers in Bi_2Te_3 across the van der Waals gap is a relatively easy process, leading to a wide dislocation core. (2) The SF minimum in Bi_2Te_3 is very shallow, 10 mJ/m^2 (compare with 100 mJ/m^2 in Cu). It is due to a combination of these two factors that the dislocation content is spread over a wide region but does not localize into discrete partial dislocations like in Cu, despite the existence of a stable SF. The latter conclusion is contrary to the assumption in the previous literature.

Reviewer:

(2) The introduction can provide more context about the impact of knowing the dislocation core structure. It is stated that dislocations can affect the functional properties of these important thermoelectric materials, however a step is missing to relate the core geometry to the functionalities. In my opinion, this should be emphasized for the Nature Communications audience.

Response:

We agree with the Reviewer that the link between the dislocation structure and the functional properties is important. Several sentences with new literature references have been added in the beginning of the third paragraph of the paper to emphasize this point. Further, the first two paragraphs of the paper have been rewritten to discuss this link in more detail. We cite recent experimental papers showing that dislocation arrays in Bi-Sb-Te alloys improve thermoelectric performance by effectively scattering mid-frequency phonons. Phonon scattering on a dislocation depends on the dislocation core structure and is shown to be an important factor in thermal conductivity of the material. Furthermore, recent STM measurements have demonstrated shifts in the energy of the massless Dirac states in Bi_2Se_3 thin films containing dislocation arrays. The effect was attributed to the large strain fields near the individually resolved dislocations cores. This was confirmed by first-principles calculations of the strain effect on the electronic band structure. Tensile strain was found to shift the Dirac point E_{D} down, while compressive strain opened a band gap and destroyed the Dirac states. In a more recent paper, experiments and first-principles calculations have shown that application of tensile and compressive stresses across the van der Waals gap in Bi_2Se_3 thin films strongly affects the electronic properties, including the work function, Fermi level and the topological Dirac states. This observation is very important because alternating tension-compression states arise near dislocation cores and thus strongly depend on the core structure. We also cite recent work on the effect of dislocations on the carrier concentrations and thus the electrical resistivity and thermopower. Furthermore, our own DFT calculations of the band structure and charge density in the SF region are reported in the paper (Supplementary Information file). We believe that the modified version of the paper does provide the context related to the impact of the dislocation structure on functional properties.

Reviewer #3:

Reviewer:

However, I am unable to conclude that the present manuscript offers the sort of truly novel or unexpected findings that would likely to excite the immediate interest of a broad range of researchers. Unfortunately, it is currently not clear to me that it fits the criteria for publication in Nature Communications. These findings might be better suited for publication in a more specialized journal. I have a few technical questions and comments listed below, but my main concern for publication in Nature Communications is the lack of evidences that the manuscript will of interest to a broad research community.

Response:

We respectfully disagree with the Reviewer on this point. Historically, the basal plane dislocations in Bi_2Te_3 were one of the first dislocations ever to be directly observed by electron microscope. Yet, their atomic-level structure has remained elusive until this paper. We have been able to observe this structure and quantify the disregistry in the dislocation core region. Contrary to the prior belief, the core is widely spread not by dissociation into partials but because of the relatively low barrier for sliding across the van der Waals gap. The particular material studied here represents a large class of quasi-2D chalcogenides that have recently attracted a great deal of attention as thermoelectric materials and topological insulators. As discussed in the paper, dislocations in such materials can strongly impact their structural and functional properties. The new understanding of the dislocation core structure achieved in this work can impact our ability to design and control the functional properties and will certainly present great interest to a broad research community.

Reviewer:

1. The authors mention that “the large strain field near dislocation cores can also affect the electronic band structure.” Have the authors investigated that with their DFT study? Can they provide any insight on that? Band structure calculations are fundamental to predict thermoelectric properties, which the authors mentioned as one of the main applications where their work can impact.

Response:

The effect of large strains near the dislocation on the band structure was not studied in this work. In fact, direct DFT simulations of wide dislocation cores are not feasible at the present time due to computational limitations of the method (see also below). However, in the first paragraph of the paper we refer to previous work in which the strain effect on electronic band structure was studied by experiments and first-principles calculations for thin films subject under tensile or compressive strains. See also our response to the second question of Reviewer #2.

Reviewer:

2. The authors mention that “the local structural disruption imposed by a dislocation presents us with an opportunity to probe the interaction strength across the interlayer gap.” Can the authors provide any insight about the interaction strength in Bi_2Te_3 from their results?

Response:

As summarized in the last paragraph of the paper, the interactions across the interlayer gap lead to a combination of two unusual features of the dislocations in Bi_2Te_3 : (1) The relatively low barrier for sliding of the layers past each other. This barrier is significantly lower than in most metals, which leads to a widely spread core that is likely to glide along the interlayer gap more easily. (2) Despite the existence of a stable SF, the energy barrier surrounding the SF energy minimum is anomalously shallow, preventing the core dissociation into partials. The core remains

wide but undissociated (that is, it does not form two distinct partial dislocations with an intervening stacking fault). These unusual features of the dislocation core structure are not found in metallic systems and owe their origin to the unique nature of the van der Waals interactions across the gap.

Reviewer:

3. How did the authors create dislocations in their DFT configurations? Did they relax the sample? Do the final results depend on the relaxation method used?

Response:

We did not create a dislocation in the DFT calculations. Given the widely dissociated core, the system containing a dislocation and its strain field would contain a large number of atoms, which is beyond the present-day capabilities of DFT calculations. This is exactly why we applied the hierarchical multi-scale approach, in which the DFT calculations were used to compute the lattice parameter, elastic constants and the gamma-surface on the basal plane, and this information was fed into the semi-discrete Peierls-Nabarro model. The latter then predicted the dislocation core structure, which was found to be in excellent agreement with experiment.

Reviewer:

4. Did any input from the experimental data help with the theoretical modeling? It is not clear from the manuscript.

Response:

The experimental observations have shown that the dislocation core terminates at the van der Waals gap, i.e., at the $\text{Te}^{(1)}\text{-Te}^{(1)}$ plane. This information was used in choosing the crystal planes on which to calculate the gamma surface. Other than this, the modeling part of the paper did not use any experimental input. The semi-discrete Peierls-Nabarro model was solely informed by DFT calculations performed in this work. This was stated in the paper, but we have slightly modified the text to emphasize this point.

Reviewer:

5. The manuscript is written assuming that the readers are familiar to the notations used, and not for a broad audience. The authors should include a brief description of the terminology they used in the main manuscript, to improve readability.

Response:

We agree that readability of the paper by a broader audience is important, especially for a Nature Communications publication. We have included several comments reminding the reader about some of the basic concepts and terminology related to dislocations, such as the Burgers vector, Burgers circuit and stacking fault.

Reviewer:

6. Why did the authors choose a polycrystalline sample? Could grain boundaries have an impact on their results?

Response:

Bulk Bi_2Te_3 -based thermoelectrics devices typically employ polycrystalline material. The observations presented here were conducted in conjunction with a study of consolidation of polycrystalline Bi_2Te_3 via hot-extrusion (as discussed in the Methods section). This approach was convenient for the present study both because the resulting deformation microstructure gives a large population of dislocations and because the preferential crystallographic texture of the

material gives rise to $\langle 2 -1 -1 0 \rangle$ -oriented grains, which is an ideal imaging direction for atomically resolving the dislocation core and its displacements. To minimize the influence of grain boundaries and other crystallographic defects on the dislocation core structures, we took care to analyze dislocations located in the central, single crystalline regions of the grains and well separated from grain boundaries and other dislocations by a radius of at least 75 nm, which is large compared with the measured core widths. All this information has been included in the revised manuscript.

Reviewer:

7. How did they initiate the dislocation in the experimental sample? Is it present in the sample or the authors applied some shear? How likely it is that these dislocations occur in samples fabricated with advanced nanofabrication techniques, e.g. MBE? No such discussions are provided.

Response:

As noted in the Methods section and above, the investigated material was processed using a thermomechanical method (hot-extrusion) and we expect that such deformation processing is the source of the dislocations in this material. Thermomechanical processing methods, including extrusion and hot-pressing, are widely employed for consolidating bulk thermoelectric materials, including Bi_2Te_3 , making the investigation of dislocations in such materials extremely relevant. Because of the importance of this point, we have added some mention of thermomechanical methods to the introductory paragraph of the paper.

Concerning the second point, we also expect dislocations to arise in the context of more advanced thin film/nanofabrication techniques such as MBE. For instance, threading dislocations have been reported in thin film chalcogenides. Dislocations can also play a role in accommodating crystallographic misfit strains and thermal mismatch strains, which is important for devices such as thermoelectrics that see large and variable temperature gradients.

Reviewer:

8. The authors mentioned that the results from their study can be generalized to understand other layered materials, but they did not provide any discussion. Without this kind of insight, unfortunately, the study seems too narrow in scope.

Response:

The particular material studied here belongs to a larger class of quasi-2D materials in which stacks of covalently-bonded atomic multilayers/sheets are separated by van der Waals gaps. Although we have studied the unique features of basal-plane dislocations in one of such materials, the common nature of the interlayer interactions strongly suggests that these features are shared by many other materials of this class. In a recent paper by Fu et al. [34], the cores of basal dislocations in another tetradymite-structured chalcogenide, Sb_2Te_3 , were also found to be localized at the van der Waals gap (even though the authors did not measure the disregistry function or analyze the gamma surface). We thus believe that that the results presented here are rather general.

REVIEWERS' COMMENTS:

Reviewer #1 (Remarks to the Author):

I agree with the author's response that one of the raised points, in particular #1, is out of the scope for the present claim. The added calculation results are informative for a further study to consider the strain effect on thermoelectric properties of Bi-Te systems. I think that the revised manuscript is now ready to be published.

Reviewer #2 (Remarks to the Author):

The authors have addressed all the comments.

Reviewer #3 (Remarks to the Author):

The authors have replied satisfactorily to all my previous questions and comments. I would appreciate if the authors make two minor modifications/additions to the manuscript:

1. It would be helpful to include a definition of disregistry in the main document. The authors might want to include the equation provided in the Methods, with clearly defining the variables, u , y and m .
2. It would be very useful if the authors can provide a cartoon sketch of the model, described beginning of Supplementary Note 1. It is hard to visualize the geometry with just words. It would be helpful for the readers if it is added to one of the main figures of the manuscript, if space is not a concern.

I recommend the manuscript for publication in Nature Communications.

Response to Reviewers' comments

We are grateful to Reviewers #1 and #2 for approving the revised version of the paper, and to Reviewer #3 for helpful comments. We are resubmitting the manuscript with changes suggested by Reviewer #3, which are highlighted in red. We have also reformatted the manuscript to match the style of *Nature Communications*.

Reviewer #3

Comment:

1. It would be helpful to include a definition of disregistry in the main document. The authors might want to include the equation provided in the Methods, with clearly defining the variables, u , y and m .

Response:

We have followed this suggestion and included a couple of additional sentences on page 5 that provide details related to the definition of the disregistry. We were unable to move the equation from the Methods section to this place, because that equation is specific to the particular orientation of the coordinate axes and crystallographic directions that are introduced later in the paper. At this point we only provide a general definition of the disregistry. We hope, however, that the additional sentences will help the reader better understand what the disregistry is.

Comment:

2. It would be very useful if the authors can provide a cartoon sketch of the model, described beginning of Supplementary Note 1. It is hard to visualize the geometry with just words. It would be helpful for the readers if it is added to one of the main figures of the manuscript, if space is not a concern.

Response:

We have followed this suggestion as well. A new cartoon sketch has been added as Figure 1c, which depicts the geometry of the dislocation in the basal plane. We hope this will help the readers visualize the dislocation line orientation, the Burgers vector, the Cartesian axes, and the relevant crystallographic directions.